# HEART-ViT: Hessian-guided Efficient Dynamic Attention and Token Pruning in Vision Transformers

## Abstract

Vision Transformers (ViTs) deliver state-of-the-art accuracy but their quadratic attention cost and redundant computations severely hinder deployment on latency- and resource-constrained platforms. Existing pruning approaches treat either tokens or heads in isolation, relying on heuristics or first-order signals, which often sacrifice accuracy or fail to generalize across inputs. We introduce HEART-ViT, a Hessian-guided efficient dynamic attention and token pruning for vision transformers, which to the best of our knowledge, is the first unified, second-order, input-adaptive framework for ViT optimization. HEART-ViT estimates curvature-weighted sensitivities of both tokens and attention heads using efficient Hessian–vector products, enabling principled pruning decisions under explicit loss budgets. This dual-view sensitivity reveals an important structural insight: token pruning dominates computational savings, while head pruning provides fine-grained redundancy removal, and their combination achieves a superior trade-off. On ImageNet-100 and ImageNet-1K with ViT-B/16 and DeiT-B/16, HEART-ViT achieves up to 49.4% FLOPs reduction, 36% lower latency, and 46% higher throughput, while consistently matching or even surpassing baseline accuracy after fine-tuning (e.g., +4.7% recovery at 40% token pruning). Beyond theoretical benchmarks, we deploy HEART-ViT on different edge devices, like-AGX Orin, demonstrating that our reductions in FLOPs and latency translate directly into real-world gains in inference speed and energy efficiency. HEART-ViT bridges the gap between theory and practice, delivering the first unified, curvature-driven pruning framework that is both accuracy-preserving and edge-efficient.

## 1 Introduction

Vision Transformers (ViTs) have rapidly become a foundation model in computer vision, achieving state-of-the-art performance across classification, detection, and segmentation tasks. Their success stems from their flexibility: ViTs can model long-range dependencies and scale effectively with data and compute. However, this power comes at a cost. Standard ViTs process hundreds of tokens through dozens of attention heads, resulting in substantial inference latency and memory footprint. Such overheads limit their adoption in real-time and resource-constrained settings, such as mobile devices or edge platforms.

A natural way to address this challenge is pruning: removing parts of the model that contribute little to performance. In convolutional networks, pruning strategies are well studied, ranging from magnitude-based weight removal to structured filter pruning. In contrast, pruning in ViTs remains less mature. Existing dynamic ViT methods typically rely on heuristics—such as entropy-based token dropping Rao et al. (2021); Zhou et al. (2025), saliency or Hessian-aware importance scores Yang et al. (2023b); Wang et al. (2021), or auxiliary gating modules for token retention Rao et al. (2021); Pan et al. (2022)—to decide what to prune. While effective in some cases, these heuristics are not explicitly tied to the model's loss, making pruning decisions less principled and sometimes unstable across inputs.

In this work, we propose **HEART-ViT** (Head and Token–Aware Pruning), a framework that performs *input-adaptive* pruning of both tokens and attention heads based on second-order sensitivity

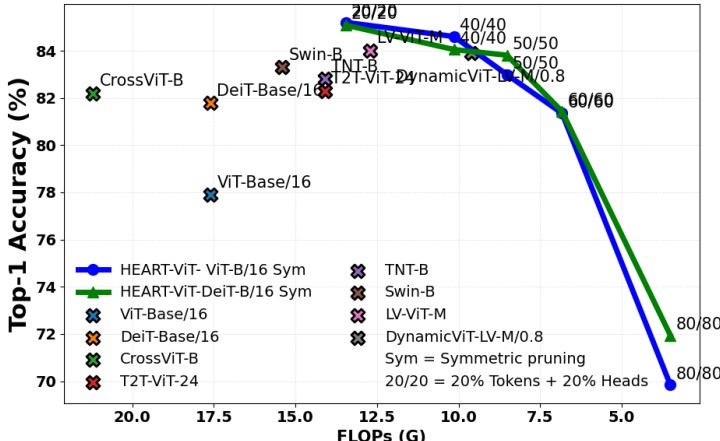

Figure 1: **Our method pushes the Pareto frontier of FLOPs vs. accuracy on ImageNet-1K.** We compare ViT-B/16 and DeiT-B/16 under symmetric (Sym) and asymmetric (Asym) pruning against strong baselines and state-of-the-art transformer variants. Our pruned models consistently achieve higher accuracy at significantly reduced FLOPs, surpassing both dense ViT/DeiT baselines and competitive efficient transformers. Detailed results are on Appendix.Table 6. *Notes: Sym = Symmetric pruning; 20/20 = 20% Tokens + 20% Heads.*

analysis. Our key insight is that the importance of a component can be quantified by how much the loss would increase if that component were removed. Using a second-order Taylor expansion, we derive a simple and elegant score: the quadratic form $z^\top \mathcal{H}_z z$, where $z$ is a token or head activation and $\mathcal{H}_z$ is the Hessian of the loss with respect to $z$. This score has two desirable properties: (i) it is *loss-aware*, directly reflecting the training objective, and (ii) it is *input-specific*, capturing which tokens and heads matter for each example.

Building on this sensitivity measure, HEART-ViT introduces a unified pruning strategy that dynamically gates tokens and heads during inference. Low-sensitivity components are pruned away, while important ones are retained. Our method supports both *symmetric pruning* (uniform ratios across layers) and *asymmetric pruning* (adaptive ratios guided by sensitivity), enabling flexible trade-offs between efficiency and accuracy. To further stabilize pruning, we employ layerwise normalization and optional soft gates, which provide a differentiable relaxation useful for fine-tuning.

Our contributions are threefold:

- We develop a principled, second-order sensitivity framework for dynamic token and head pruning in Vision Transformers.
- We unify token and head pruning under the same importance criterion, avoiding the need for heuristic rules or auxiliary predictors.
- We demonstrate that HEART-ViT achieves significant FLOPs and latency reductions while preserving, and in some cases improving, accuracy on challenging benchmarks.

HEART-ViT brings a loss-aware, mathematically grounded perspective to pruning in Vision Transformers, bridging the gap between theory and practice in efficient transformer inference. To highlight our contributions in context, Table 1 presents a comparative novelty timeline (2021–2025), situating HEART-ViT relative to representative ViT pruning methods. This timeline illustrates how HEART-ViT is the first to combine *second-order sensitivity*, *unified token & head pruning*, and *explicit loss-budget control*, going beyond prior heuristic- or module-based approaches.

## 2 RELATED WORK

Transformers have revolutionized computer vision and NLP tasks but remain computationally expensive for deployment in resource-constrained settings. Numerous studies have proposed techniques to prune redundant components while preserving accuracy. Early efforts like Weight Magnitude Pruning and Structured Pruning focused on removing entire layers or heads based on fixed heuristics Molchanov et al. (2017b). Later, Rao et al. (2021) and Chavan et al. (2022) introduced dynamic token pruning based on learned importance scores, reducing inference costs by skipping unimportant tokens.

| Year | 2021a | 2021b | 2022a | 2022b | 2023 | 2024 | 2025 (Ours) |
|---|---|---|---|---|---|---|---|
| Method | TokenLearner | DynamicViT | EViT | SPViT | ToMe | Sparse-then-Prune | **HEART-ViT** |
| | Ryoo et al. (2021) | Rao et al. (2021) | Liang et al. (2022) | Kong et al. (2022) | Bolya et al. (2023b) | Prasetyo et al. (2023) | (Ours) |
| **Core Techniques** | | | | | | | |
| Token pruning | ✓ | ✓ | ✓ | ✓ | ✓ | ✓ | ✓ |
| Head pruning | ✗ | ✗ | ✗ | ✗ | ✗ | ✓(structural) | ✓(**dynamic**) |
| Attention/CLS-based heuristics | ✗ | ✓ | ✓ | ✓ | ✓ | ✓ | ✗ |
| Learned token/module generator | ✓ | ✓ | ✗ | ✓ | ✗ | ✗ | ✗ |
| Second-order Hessian sensitivity | ✗ | ✗ | ✗ | ✗ | ✗ | ✗ | ✓ |
| Unified token & head pruning | ✗ | ✗ | ✗ | ✗ | ✗ | ✗ | ✓ |
| **Objective / Loss-level Innovations** | | | | | | | |
| Per-input dynamic pruning | ✓ | ✓ | ✓ | ✓ | ✓ | ✗ | ✓ |
| Taylor-series $\Delta\mathcal{L}$ linkage | ✗ | ✗ | ✗ | ✗ | ✗ | ✗ | ✓ |
| Loss-budget constraint ($\varepsilon$) | ✗ | ✗ | ✗ | ✗ | ✗ | ✗ | ✓ |
| Soft-to-hard gating (annealed) | ✗ | ✗ | ✗ | ✗ | ✗ | ✗ | ✓ |
| Symmetric vs. Asymmetric pruning | ✗ | ✗ | ✗ | ✗ | ✗ | ✗ | ✓ |
| **Empirical Evaluation Paradigms** | | | | | | | |
| Latency / throughput analysis | ✗ | ✓ | ✓ | ✓ | ✓ | ✗ | ✓ |
| Edge-device relevance | ✗ | ✗ | ✗ | ✓ | ✓ | ✗ | ✓ |
| Ablations across heads & tokens | ✗ | ✗ | ✗ | ✗ | ✗ | ✗ | ✓ |

Table 1: Comparative novelty timeline (2021–2025) situating our **HEART-ViT** framework relative to representative ViT pruning approaches. Unlike prior heuristics or static structural methods, HEART-ViT introduces a *unified, second-order, loss-aware* pruning criterion covering both tokens and heads, and explores *symmetric vs. asymmetric* pruning strategies under explicit loss budgets.

Recent studies such as Yang et al. (2023a) expand this line of work by introducing layer-agnostic Hessian-based saliency scores and latency-aware regularization to improve deployability on edge devices. Token pruning reduces the input sequence length during inference, targeting redundant spatial information. Bolya et al. (2023a) proposed merging similar tokens via bipartite matching, while Ruan et al. (2021) employed gating mechanisms to prune tokens dynamically per sample. However, these methods often rely on learned importance scores or attention weights, which may not correlate well with actual sensitivity to loss.

More recently, methods like Fu et al. (2024) and Tao et al. (2025) introduced efficient, training-free or lightweight strategies for dynamic pruning in long-context LLMs and ViTs, showing that token-level adaptivity can be achieved without sacrificing performance.

Attention head pruning aims to remove redundant heads within multi-head attention (MHA). While earlier works used attention entropy Michel et al. (2019) or magnitude-based heuristics, such approaches fail to consider second-order sensitivity. Taylor expansion-based head pruning Sanh et al. (2020) improved interpretability but remained static across inputs. Recent techniques, such as Entropy-Guided Head Importance Lee & s. Kim (2024), leverage entropy metrics to guide pruning and mitigate information loss. Additionally, Automatic Channel Pruning for Multi-Head Attention Lee & Hwang (2024) tackles head-level redundancy through channel similarity-based heuristics.

Hessian-based methods evaluate the impact of pruning on loss via second-order derivatives, offering a principled criterion for parameter importance. Notable early works such as Molchanov et al. (2017a) and Singh et al. (2020)applied Hessian approximations to CNN pruning. In ViTs, Yu et al. (2022) extended this concept to structured attention pruning.

More recently, SwiftPrune Kang et al. (2025) introduced a Hessian-free metric for LLM weight pruning, while SNOWS Lucas & Mazumder (2024) proposed a second-order optimization framework without explicit Hessian computation, targeting global objective preservation. These methods aim to reduce computational overhead while maintaining the benefits of second-order sensitivity.

Recent advances focus on input-conditional computation. AdaViT Mullapudi et al. (2022) and DeiT-GATE Tang et al. (2023) introduced gating mechanisms to prune tokens and heads based on input content. However, these methods often require reinforcement learning or complex routing logic, which can increase inference time and destabilize training. To address this, Automatic Pruning Rate Adjustment Ishibashi & Meng (2025) and OptiPrune Le et al. (2025) propose adaptive sparsity control via gradient-aware or loss-aware criteria for sample-dependent efficiency.

Unlike prior methods that rely on attention magnitude or learned heuristics, HEART-ViT introduces a Hessian-based dynamic pruning framework that jointly estimates token and head sensitivity through second-order approximations. By integrating Hutchinson's estimator and adaptive gating, our method performs input-dependent structured pruning, achieving competitive accuracy with lower computational cost. Furthermore, HEART-ViT bridges the gap between dynamic inference and mathematically grounded sensitivity measures, setting a new direction for efficient transformer deployment.

## 3 TECHNICAL APPROACH

### 3.1 PROBLEM SETUP

Vision Transformers (ViTs) achieve strong accuracy but incur high inference cost. Given a pre-trained ViT $f_\theta$ and a data distribution $\mathcal{D}$, our goal is *per-input* dynamic pruning of tokens and attention heads with minimal loss increase. For an input image $x$, the ViT processes a sequence of token embeddings $X^0 = [t_1^0, \ldots, t_{n_0}^0] \in \mathbb{R}^{n_0 \times d}$ produced by patch embedding plus positional encodings. Each transformer layer $\ell = 1, \ldots, L$ applies

$$X^\ell = \text{FFN}^\ell\Big(\text{MSA}^\ell(\text{LN}(X^{\ell-1}))\Big) + X^{\ell-1}.$$

Multi-head self-attention (MSA) uses $H_\ell$ heads,

$$\text{MSA}^\ell(X) = \big[\, h_1^\ell(X); \ldots; h_{H_\ell}^\ell(X)\,\big] W_O^\ell, \quad h_k^\ell(X) = \text{softmax}\Big(\frac{Q_k K_k^\top}{\sqrt{d_k}}\Big) V_k,$$

with $Q_k = X W_k^{\ell,Q}$, $K_k = X W_k^{\ell,K}$, $V_k = X W_k^{\ell,V}$.

We introduce binary masks over *token activations* and *head outputs* at inference:

$$M_T^\ell \in \{0,1\}^{n_\ell}, \qquad M_A^\ell \in \{0,1\}^{H_\ell}.$$

Let $g_T^\ell = \text{diag}(M_T^\ell)$ and $g_A^\ell = \text{diag}(M_A^\ell)$ denote token and head gates. Gated forward pass applies

$$\tilde{X}^{\ell-1} = g_T^{\ell-1} X^{\ell-1}, \qquad \widetilde{\text{MSA}}^\ell(\tilde{X}^{\ell-1}) = \big[\, g_A^\ell h_1^\ell(\tilde{X}^{\ell-1}); \ldots; g_A^\ell h_{H_\ell}^\ell(\tilde{X}^{\ell-1})\,\big] W_O^\ell,$$

and proceeds as usual. The prediction for input $x$ under masks $M = \{M_T^\ell, M_A^\ell\}_{\ell=1}^L$ is $f_\theta(x; M)$.

**Objective.** We seek sparse masks that preserve accuracy in expectation:

$$\min_M \ \mathbb{E}_{(x,y)\sim\mathcal{D}} \, \mathcal{L}\big(y, f_\theta(x; M)\big) \ + \ \lambda_T \sum_\ell \|M_T^\ell\|_0 \ + \ \lambda_A \sum_\ell \|M_A^\ell\|_0. \qquad (1)$$

Equivalently, for a loss budget $\varepsilon > 0$ we may constrain

$$\min_M \ \sum_\ell \Big( \|M_T^\ell\|_0 + \|M_A^\ell\|_0 \Big) \quad \text{s.t.} \quad \mathbb{E}_{(x,y)}\Big[\Delta\mathcal{L}(x; M)\Big] \leq \varepsilon, \qquad (2)$$

with $\Delta\mathcal{L}(x; M) = \mathcal{L}(y, f_\theta(x; M)) - \mathcal{L}(y, f_\theta(x; \mathbf{1})) \geq 0$.

Eqs. 1–2 form a dual perspective: one emphasizes sparsity with loss regularization, the other enforces an explicit error budget. This flexibility allows HEART-ViT to be deployed under accuracy constraints (e.g., real-time inference) or resource constraints (e.g., FLOPs budgets).

### 3.2 SECOND-ORDER, INPUT-AWARE SENSITIVITY

For dynamic pruning we score, *per input* $x$, the loss sensitivity to removing a component $z$ (a token activation $t_j^\ell(x)$ or a head output $h_k^\ell(x)$). Consider perturbing the component by $\Delta z$ and apply a second-order Taylor expansion around the converged model:

$$\mathcal{L}(z + \Delta z) \approx \mathcal{L}(z) + \nabla_z \mathcal{L}^\top \Delta z + \tfrac{1}{2} \Delta z^\top \underbrace{\nabla_z^2 \mathcal{L}}_{\mathcal{H}_z} \Delta z.$$

Substituting $\Delta z = -z$ gives

$$\Delta\mathcal{L}_z(x) \approx -\nabla_z \mathcal{L}^\top z(x) + \tfrac{1}{2} z(x)^\top \mathcal{H}_z(x) z(x).$$

At convergence $\nabla_z \mathcal{L} \approx 0$, leaving only the curvature term:

$$\Delta\mathcal{L}_z(x) \approx \tfrac{1}{2} z(x)^\top \mathcal{H}_z(x) z(x). \qquad (3)$$

The loss increase from removing a component is controlled by the curvature along its activation direction: flatter directions cause little change, while steeper ones are costly to prune.
We therefore define the *second-order importance* (up to a factor of 2):

$$S_z(x) = z(x)^\top \mathcal{H}_z(x) z(x), \qquad z \in \big\{\, t_j^\ell(x), h_k^\ell(x) \,\big\}. \qquad (4)$$

$S_z$ measures how much curvature-weighted energy a token or head carries toward the loss; pruning low-$S_z$ elements discards directions of minimal influence. Both token and head scores are activation-based and thus input-adaptive.

**Efficient evaluation via HVP.** We never form $\mathcal{H}_z$ explicitly. Using Pearlmutter's trick,

$$\mathcal{H}_z(x)\,z(x) \;=\; \frac{d}{d\epsilon}\,\nabla_z \mathcal{L}\big(z(x) + \epsilon z(x)\big)\Big|_{\epsilon=0},$$

so $S_z(x) = \langle \mathcal{H}_z(x)z(x),\, z(x)\rangle$ is computed with two backprop passes. For robustness we average over a small calibration batch $\mathcal{B}$:

$$\bar{S}_z \;=\; \frac{1}{|\mathcal{B}|} \sum_{(x,y)\in\mathcal{B}} S_z(x). \tag{5}$$

This averaging stabilizes scores, ensuring that pruning decisions reflect consistent importance across inputs rather than noise from a single example.

**Safe pruning budget.** If we prune a set $\mathcal{Z}$, the total loss increase is additively approximated:

$$\Delta\mathcal{L}(x; \mathcal{Z}) \;\approx\; \frac{1}{2}\sum_{z\in\mathcal{Z}} S_z(x).$$

Enforcing $\sum_{z\in\mathcal{Z}} S_z(x) \leq 2\varepsilon$ ensures $\Delta\mathcal{L} \leq \varepsilon$ under the quadratic model.

**Hutchinson's Estimator.** The quadratic form in Eq. 4 can be interpreted as a rank-1 trace. Recall Hutchinson's identity for any symmetric matrix $H$:

$$\mathrm{Tr}(H) \;=\; \mathbb{E}_{v\sim\mathrm{Rad}(\pm 1)}\big[v^\top H v\big].$$

If we restrict the trace to the one-dimensional subspace spanned by $z(x)$,

$$S_z(x) = \|z(x)\|_2^2 \cdot \frac{z(x)}{\|z(x)\|_2}^\top \mathcal{H}_z(x)\,\frac{z(x)}{\|z(x)\|_2},$$

we obtain the Rayleigh quotient of $\mathcal{H}_z$ along $z(x)$. Thus $S_z$ is a principled specialization of Hutchinson-type estimators with a deterministic probe vector, situating HEART-ViT in a broader stochastic-estimation framework.

### 3.3 UNIFIED TOKEN & HEAD SCORING

**Token sensitivity.** For token $t_j^\ell(x) \in \mathbb{R}^d$,

$$S_{t_j^\ell}(x) \;=\; t_j^\ell(x)^\top \mathcal{H}_{t_j^\ell}(x)\, t_j^\ell(x). \tag{6}$$

Aggregating across layers yields $S_{t_j}^{\mathrm{agg}}(x) = \sum_\ell S_{t_j^\ell}(x)$.

**Head sensitivity.** For head output $h_k^\ell(x) \in \mathbb{R}^{n_\ell \times d_k}$, we score its vectorization:

$$S_{a_k^\ell}(x) \;=\; \mathrm{vec}\big(h_k^\ell(x)\big)^\top \mathcal{H}_{h_k^\ell}(x)\, \mathrm{vec}\big(h_k^\ell(x)\big). \tag{7}$$

Thus tokens and heads share the same curvature-weighted quadratic form, differing only in dimensionality. This symmetry highlights that HEART-ViT unifies representational (tokens) and operational (heads) pruning under one criterion.

**Normalization and ranking.** Within each layer $\ell$ we normalize scores:

$$\hat{S}_z \;=\; \frac{S_z - \mu_\ell}{\sigma_\ell}, \qquad \mu_\ell = \mathrm{mean}\{S_z\}_{z\in\mathcal{C}_\ell}, \;\; \sigma_\ell = \mathrm{std}\{S_z\}_{z\in\mathcal{C}_\ell}, \tag{8}$$

where $\mathcal{C}_\ell$ is the candidate set. Components are pruned by threshold or percentile.

### 3.4 DYNAMIC PRUNING POLICY

For input $x$, HEART-ViT computes $\{S_z(x)\}$ per layer and applies either hard or soft gating:

**Hard selection (inference).** Keep top-$k$ by $S_z$ or prune those with $S_z < \tau$:

$$M_z(x) \;=\; \mathbb{I}\big[S_z(x) \geq \tau\big]. \tag{9}$$

**Soft gating (fine-tuning).** Use a sigmoid gate with annealed $\gamma$ so $G_z$ approaches a hard mask as:

$$G_z(x) \;=\; \sigma\big(\gamma(\hat{S}_z(x) - \tau)\big) \in (0,1), \tag{10}$$

Forward uses $G_z$ multiplicatively on tokens and head outputs.

Hard gating guarantees strict sparsity for deployment; soft gating is a differentiable relaxation enabling gradient-based fine-tuning. This connects HEART-ViT to continuous relaxations used in differentiable NAS, while retaining interpretability as second-order loss-aware pruning.

**Complexity.** Per layer, sensitivity evaluation uses one forward and two backprops for HVP; selection is $O(n_\ell \log n_\ell)$ for $n_\ell$ candidates (tokens+heads). Calibration amortizes over many inputs ($|\mathcal{B}| \ll |\mathcal{D}|$). Overhead is small relative to the ViT forward, while pruning reduces FLOPs/latency.

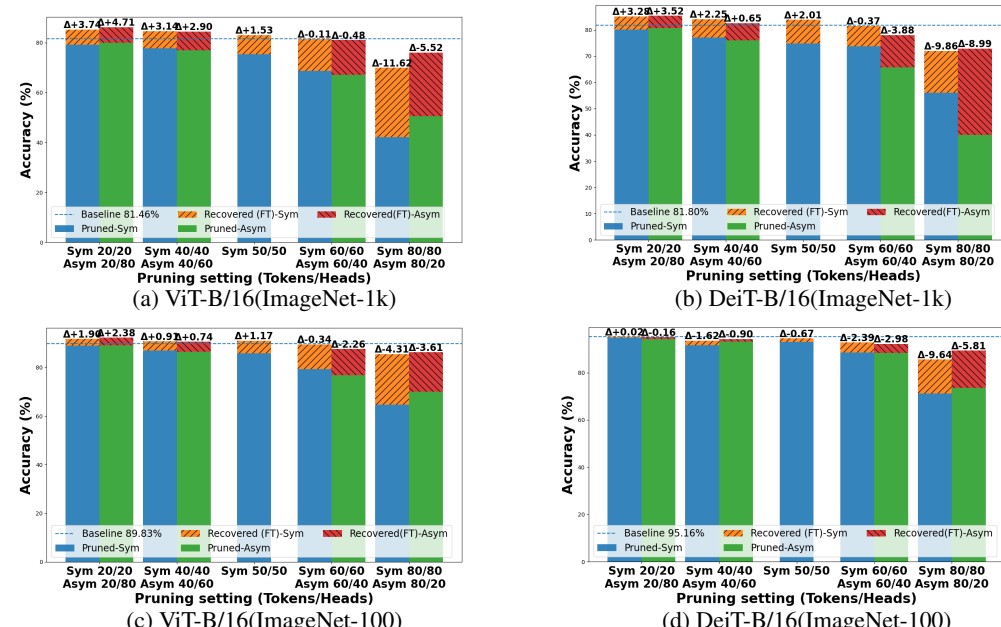

Figure 2: Accuracy decomposition under symmetric (Sym) and asymmetric (Asym) pruning. (a)–(b) show results on ImageNet-1K for ViT-B/16 and DeiT-B/16, while (c)–(d) present the corresponding results on ImageNet-100. Bars indicate pruned retention (bottom) and accuracy recovered by fine-tuning (FT, top); the dashed line marks the dense baseline; Δ annotations indicate the accuracy change relative to the baseline.detailed results has been shown in (Appendix: Table 2 & 3.)

## 4 EXPERIMENT

### 4.1 EXPERIMENTAL SETUP

We conduct experiments on ImageNet-1K and ImageNet-100 using ViT-B/16 and DeiT-B/16 backbones. HEART-ViT is evaluated under both symmetric and asymmetric pruning ratios, with pre- and post-fine-tuning results reported. A detailed description of datasets, architectures, training hyperparameters, and pruning configurations is provided in Appendix A.1.

### 4.2 RESULT

Figure 3 - (a)(b) (ImageNet-1k) & 3(b)(c) (ImageNet-100) present the results of symmetric and asymmetric token & head pruning using HEART-ViT on ViT-B/16 and DeiT-B/16.(detailed results in Appendix. Table 2 and 3 ) In symmetric pruning, tokens and heads are removed at the same rate (e.g., 50%/50%), while in asymmetric pruning, they are pruned at different rates (e.g., 20% tokens / 80% heads). This distinction is important because tokens and heads play different roles in representational capacity: pruning them uniformly may discard essential information, whereas asymmetric pruning leverages the higher redundancy of attention heads to achieve better efficiency–accuracy trade-offs.

Our pruning method in ViT-B/16 with ImageNet-100, shows substantial efficiency gains with strong accuracy recovery after fine-tuning. At 50% symmetric pruning, FLOPs are reduced by half while final accuracy improves to 91.0% (+1.17% over baseline). Even stronger gains are achieved with 20/80 asymmetric pruning, which delivers 92.21% final accuracy, the best result on this dataset (+2.38% over baseline). DeiT-B/16 shows similar resilience: 20/20 symmetric pruning achieves 95.18%, matching the dense model, while 20/80 asymmetric pruning slightly improves to 95.0%, confirming the robustness of distilled supervision under pruning.

Also in ImageNet-1K, the same trends are observed at a larger scale. For ViT-B/16, 20/80 asymmetric pruning achieves 86.17% final accuracy, outperforming the dense baseline by +4.71%, while symmetric 50% pruning reduces FLOPs by half and still maintains 82.99% accuracy. For DeiT-B/16, both strategies surpass baseline at moderate pruning: 20/20 symmetric pruning yields 82.89% (+3.28% vs. baseline) and 20/80 asymmetric pruning improves further to 83.13% (+3.52% vs. baseline). Notably, DeiT-B/16 exhibits exceptional recovery under extreme pruning: after 80/20 pruning, accuracy recovers by +32.8% to reach 70.62%, and after 80/80 pruning, it regains +15.9% to reach 69.75%, even with a 79% FLOPs reduction.

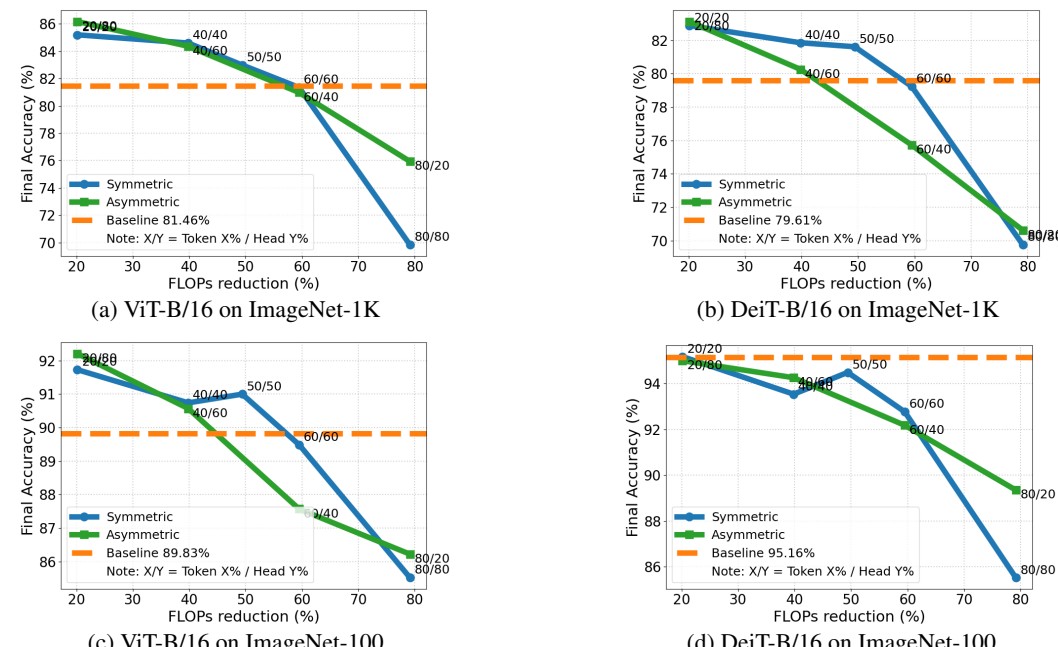

(a) ViT-B/16 on ImageNet-1K          (b) DeiT-B/16 on ImageNet-1K

(c) ViT-B/16 on ImageNet-100          (d) DeiT-B/16 on ImageNet-100

Figure 3: Accuracy vs FLOPs reduction curves for Symmetric and Asymmetric pruning on ImageNet-1K (a–b) and ImageNet-100 (c–d). Baseline accuracy is shown as dashed lines.

Observations across both ImageNet-1k and ImageNet-100, Fine-tuning shown in figure 2 is essential for recovery at higher pruning ratios ($\geq$60%), where accuracy drops sharply before adaptation. Across datasets and backbones, asymmetric pruning consistently outperforms symmetric pruning at moderate ratios, validating HEART-ViT's sensitivity-driven approach to pruning tokens and heads unevenly. DeiT-B/16 demonstrates stronger resilience under extreme pruning than ViT-B/16, suggesting that distilled supervision confers additional robustness to structural sparsification.

Both the tables 2 and 3 and figure 3 together show that HEART-ViT consistently achieves Pareto-optimal trade-offs in FLOPs, throughput, and accuracy. Symmetric pruning provides a stable baseline, while asymmetric pruning yields superior performance–efficiency trade-offs by leveraging the different redundancy levels of tokens and heads. These findings confirm that sensitivity-aware asymmetric pruning is a principled and scalable strategy for improving the efficiency of Vision Transformers across datasets and architectures.

### 4.3 LAYERWISE REPRESENTATION ANALYSIS.

We further compare ViT-B/16 and DeiT-B/16 under identical 50% token + 50% head symmetric pruning to understand how architectural and training differences influence robustness (Fig. 4 for ViT-B, Fig. 5 for DeiT-B). Across both backbones, CKA similarity decays monotonically with depth, indicating that pruning perturbs intermediate hidden states most strongly in shallow-to-mid layers. However, DeiT exhibits lower CKA values overall, reflecting greater representational shifts, likely due to its stronger distillation-driven supervision which constrains dense features more tightly and amplifies pruning perturbations.

CLS-token cosine similarity shows a similar trend: ViT's CLS trajectory diverges smoothly yet remains tightly coupled with the dense model (>0.992), while DeiT's CLS cosine drops earlier and more sharply, suggesting that DeiT reorganizes its global semantic embedding more aggressively under pruning. Residual ratios also highlight a key difference: ViT maintains a smooth transformation-to-identity tradeoff across depth, with fine-tuning restoring balance in later layers. DeiT instead shows sharper fluctuations in residual norms, with pruned layers oscillating more between identity-like and transformation-heavy behavior.

Taken together, these results reveal that ViT's representations are more stable under structured pruning, while DeiT undergoes stronger representational reshaping. Fine-tuning compensates in both cases, but the trajectory differs: ViT converges to a nearby solution space, whereas DeiT explores a more reorganized representational geometry. Extended analyses, including more pruning ratios and asymmetric settings, are provided in the Appendix.

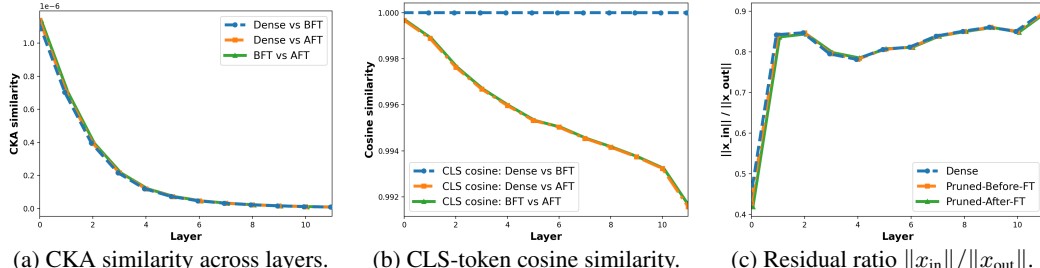

(a) CKA similarity across layers.  (b) CLS-token cosine similarity.  (c) Residual ratio $\|x_{\text{in}}\|/\|x_{\text{out}}\|$.

Figure 4: Layerwise analysis of ViT-B/16 under **50% symmetric pruning** (tokens + heads) on ImageNet-1K. CKA reveals mid-layer representational shifts that are partially recovered after fine-tuning. CLS cosine shows pruning drives the model toward an alternative semantic trajectory, while residual ratios highlight temporary suppression of transformations that fine-tuning restores.

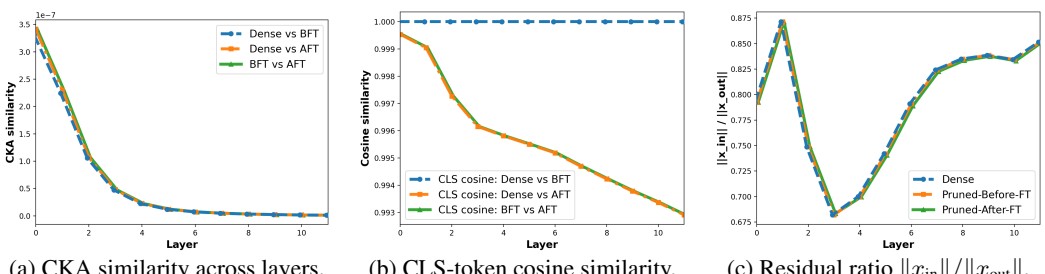

(a) CKA similarity across layers.  (b) CLS-token cosine similarity.  (c) Residual ratio $\|x_{\text{in}}\|/\|x_{\text{out}}\|$.

Figure 5: Layerwise analysis of DeiT-B/16 under **50% symmetric pruning** on ImageNet-1K. Compared to ViT, DeiT shows stronger representational shifts in CKA and CLS similarity, and larger fluctuations in residual ratios, indicating that its distilled training makes it more sensitive to pruning perturbations.

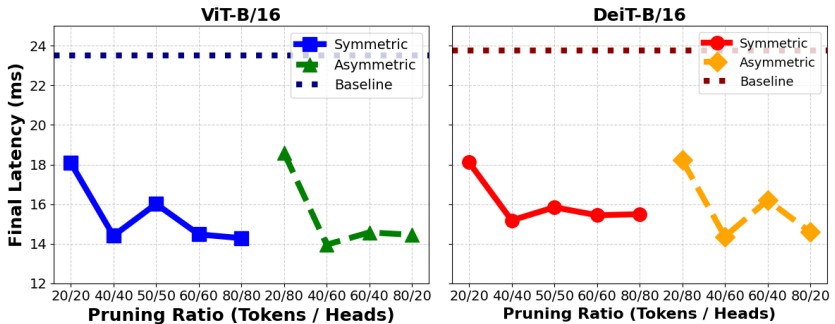

Figure 6: **Jetson Orin AGX latency vs. pruning ratio on ImageNet-1K for ViT-B/16 and DeiT-B/16.** HEART-ViT reduces latency from baseline values of ∼23 ms to ∼13–14 ms, achieving up to **40.6% improvement**. Asymmetric pruning occasionally outperforms symmetric pruning, highlighting the benefit of token–head imbalance.

### 4.4 EXPERIMENTS ON EDGE DEVICES

To evaluate deployment efficiency, we conducted experiments on a range of NVIDIA Jetson Orin edge devices with varying computational capabilities (Appendix. Table 7). These include AGX Orin, Orin NX, and Orin Nano variants, covering GPU core counts from 512 to 2048 and memory capacities from 4 GB to 32 GB. Due to space constraints, we present detailed results for the AGX Orin device, while noting that similar trends were consistently observed across the other platforms.

On AGX Orin, HEART-ViT achieves substantial latency savings for both ImageNet-1K (fig. 6) and ImageNet-100 (fig. 7). Baseline inference latencies of 23 ms are reduced to 13–14 ms at moderate pruning ratios (40–60%), corresponding to 35–43% improvements. Symmetric pruning yields smooth, monotonic reductions, while asymmetric pruning reveals sharper drops at selective configurations (e.g., 40/60 in DeiT-B/16 on ImageNet-1K, 60/40 in ViT-B/16 on ImageNet-100). These findings validate the dual-view sensitivity principle of HEART-ViT: token pruning drives bulk savings, while head pruning captures fine-grained redundancy, with asymmetric schedules often outperforming symmetric ones at equivalent budgets. Also, ViT-B/16 exhibits slightly higher maximum speedups (up to 43.1% on ImageNet-100) than DeiT-B/16, reflecting its greater redundancy compared to the distillation-optimized DeiT architecture.

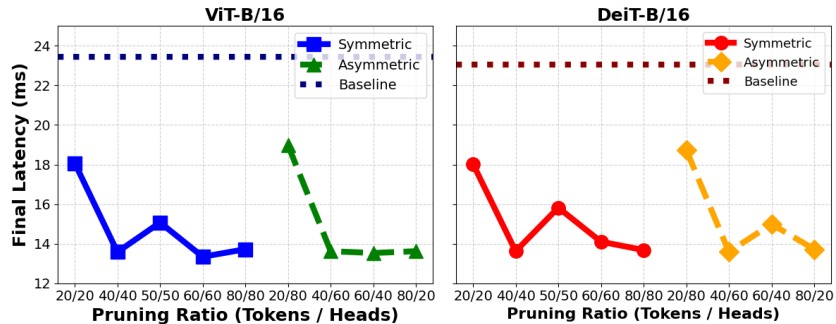

Figure 7: **Jetson Orin AGX latency vs. pruning ratio on ImageNet-100 for ViT-B/16 and DeiT-B/16.** HEART-ViT consistently achieves up to **43.1% latency reduction**, with symmetric pruning yielding smooth improvements while asymmetric pruning unlocks sharper drops at selective ratios. Similar efficiency trends were observed on other Orin devices (Appendix. Table 7).

## 5 ABLATION STUDY

Our asymmetric pruning results reveal that the efficiency improvements of Vision Transformers are overwhelmingly determined by the proportion of tokens removed, while the impact of head pruning is comparatively minor. For example, pruning $40\%$ of tokens and $60\%$ of heads reduces FLOPs by $\sim 39\text{–}40\%$, closely matching the token ratio. Conversely, pruning $80\%$ of heads but only $20\%$ of tokens reduces FLOPs by merely $\sim 20\%$. These observations indicate that FLOPs reduction consistently tracks the percentage of tokens pruned, highlighting the dominant role of token pruning in computational efficiency.

To understand this phenomenon, consider the self-attention operation at layer $\ell$ with $n_\ell$ tokens, hidden dimension $d$, and $H_\ell$ heads:

$$h_k^\ell(X) = \text{softmax}\left(\frac{Q_k K_k^\top}{\sqrt{d_k}}\right)V_k, \qquad Q_k = XW_k^{\ell,Q}, \ K_k = XW_k^{\ell,K}, \ V_k = XW_k^{\ell,V}, \qquad (11)$$

with the multi-head output

$$\text{MSA}^\ell(X) = [h_1^\ell(X); \ldots; h_{H_\ell}^\ell(X)]W_O^\ell. \qquad (12)$$

The dominant cost arises from the quadratic query–key product:

$$\text{FLOPs}^\ell \approx O(H_\ell \cdot n_\ell^2 \cdot d_k). \qquad (13)$$

Pruning tokens reduces the number of rows and columns in the attention matrix:

$$n_\ell \to \alpha n_\ell \quad \Rightarrow \quad \text{FLOPs}_{\text{tokens}}^\ell \sim O(H_\ell \cdot (\alpha n_\ell)^2 \cdot d_k), \qquad (14)$$

which produces a *quadratic* reduction in FLOPs. Where, pruning heads only decreases the number of parallel attention maps:

$$H_\ell \to \beta H_\ell \quad \Rightarrow \quad \text{FLOPs}_{\text{heads}}^\ell \sim O(\beta H_\ell \cdot n_\ell^2 \cdot d_k), \qquad (15)$$

which is only a *linear* reduction. This explains why FLOPs savings empirically align with token pruning ratios and not head pruning ratios: tokens govern the quadratic term, while heads merely scale it linearly. Curvature-based sensitivity analysis further reinforces this asymmetry. The importance of a component is measured as

$$S_z^\ell(x) = z(x)^\top \mathcal{H}_z^\ell(x)\, z(x), \qquad (16)$$

where $z(x)$ is either a token activation or a head output, and $\mathcal{H}_z^\ell$ is the Hessian restricted to that component. Token activations, especially those from background patches in early layers, often align with flat curvature directions, yielding small $S_z^\ell(x)$ and making them inexpensive to prune. Head outputs, however, correspond to specialized functional subspaces (locality, long-range context, semantics) that align with sharper curvature directions, making their removal more costly to accuracy.

Layer-wise dynamics also differ. In early layers, token redundancy is high and many background tokens can be safely removed, while retaining all heads preserves representational diversity. In middle layers, token pruning remains effective, but head pruning begins to harm feature aggregation. In later layers, tokens are already compressed toward the class token, and pruning heads disproportionately damages semantic integration. Thus, token pruning primarily governs computational efficiency, while head pruning controls representational diversity.

Overall, token pruning dominates head pruning because it reduces the quadratic computational core of self-attention, exploits the redundancy of background tokens, and directly improves hardware efficiency. Head pruning, while important for reducing parameter count and balancing accuracy, has a secondary effect on computation. This explains why asymmetric pruning strategies biased toward token pruning consistently achieve the best trade-off between efficiency and accuracy.

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

# A  APPENDIX

## A.1  EXPERIMENTAL SETUP

**Datasets.**  **ImageNet-1K** consists of 1.28M training and 50K validation images over 1,000 classes at $224\times224$ resolution. We report single-crop Top-1 accuracy on the validation set. **ImageNet-100** is a 100-class subset of ImageNet with the same resolution and evaluation protocol; we use the canonical class list and standard train/val split.

**Architectures.**  We evaluate two backbones: **ViT-B/16** (12 layers, 12 heads, patch size 16, hidden dim 768) and **DeiT-B/16** (same macro-architecture, data-efficient training). All models use $224\times224$ inputs, a class token, learned positional embeddings, and LayerNorm pre/post as in their original releases.

**Methods Compared.**  **Baseline (Dense):** the unpruned backbone trained/fine-tuned under our recipe. **HEART-ViT (ours):** Hessian-guided dynamic token+head pruning. We evaluate *Symmetric* ratios (Tokens/Heads = 20/20, 40/40, 50/50, 60/60, 80/80) and *Asymmetric* ratios (20/80, 40/60, 60/40, 80/20). We report both pre-fine-tuning and post-fine-tuning accuracy.

**Training & Fine-Tuning Protocol.**  Unless stated otherwise, we follow standard ViT/DeiT practice. *Optimizer:* AdamW ($\beta_1$=0.9, $\beta_2$=0.999), weight decay 0.05. *LR schedule:* cosine decay with 5-epoch linear warm-up; peak LR $5\times10^{-4}$ for ViT-B/16 and $3\times10^{-4}$ for DeiT-B/16 (scaled with global batch size when applicable). *Batching:* global batch size 1024 (with gradient accumulation if needed); mixed precision (AMP). *Augmentation:* RandAugment, random resized crop to 224, horizontal flip; label smoothing $\varepsilon$=0.1; Mixup= 0.8 and CutMix= 1.0 during training (disabled for evaluation). *Regularization:* DropPath 0.1 for ViT-B/16 and 0.1–0.2 for DeiT-B/16. *Initial training:* 200 epochs with early exit (patience = 15 epochs), monitoring validation Top-1. *Fine-tuning after pruning:* 100 epochs with the same early-exit rule (patience = 15). *Hardware:* all experiments run on NVIDIA A100 GPUs.

| Dataset | Model | Pruning Strategy | Ratio (Tokens/Heads) | Baseline Acc. | Pruned Acc. (Before FT) | Final Acc. (After FT) | FLOPs (G) | ΔAcc. | Recovered Acc. | FLOPs ↓ | Throughput ↑ |
|---|---|---|---|---|---|---|---|---|---|---|---|
| ImageNet-100 | ViT-B/16 | Symmetric | 20% / 20% | | 88.91 | **91.73** | 13.45 | +1.90 | +2.82 | 20.17% | +17.45% |
| | | | 40% / 40% | | 86.99 | **90.74** | 10.14 | +0.91 | +3.75 | 39.83% | +21.96% |
| | | | 50% / 50% | | 85.83 | **91.00** | 8.52 | +1.17 | +5.17 | 49.40% | +25.15% |
| | | | 60% / 60% | | 79.32 | **89.49** | 6.83 | -0.34 | +10.17 | 59.49% | +35.71% |
| | | | 80% / 80% | 89.83 | 64.68 | **85.52** | 3.51 | -4.31 | +20.84 | 79.15% | +45.51% |
| | | Asymmetric | 20% / 80% | | 89.18 | **92.21** | 13.45 | +2.38 | +3.03 | 20.17% | +17.20% |
| | | | 40% / 60% | | 86.52 | **90.57** | 10.14 | +0.74 | +4.05 | 39.83% | +21.79% |
| | | | 60% / 40% | | 76.73 | **87.57** | 6.83 | -2.26 | +10.84 | 59.49% | +35.81% |
| | | | 80% / 20% | | 70.05 | **86.22** | 3.51 | -3.61 | +16.17 | 79.15% | +44.76% |
| | DeiT-B/16 | Symmetric | 20% / 20% | | 94.77 | **95.18** | 13.45 | +0.02 | +0.41 | 20.17% | +16.59% |
| | | | 40% / 40% | | 91.58 | **93.54** | 10.14 | -1.62 | +1.96 | 39.83% | +20.91% |
| | | | 50% / 50% | | 92.92 | **94.49** | 8.52 | -0.67 | +1.57 | 49.40% | +23.82% |
| | | | 60% / 60% | | 88.53 | **92.77** | 6.83 | -2.39 | +4.24 | 59.49% | +33.26% |
| | | | 80% / 80% | 95.16 | 71.18 | **85.52** | 3.51 | -9.64 | +14.34 | 79.15% | +42.77% |
| | | Asymmetric | 20% / 80% | | 94.25 | **95.00** | 13.45 | -0.16 | +0.75 | 20.17% | +16.64% |
| | | | 40% / 60% | | 93.11 | **94.26** | 10.14 | -0.90 | +1.15 | 39.83% | +21.21% |
| | | | 60% / 40% | | 88.41 | **92.18** | 6.83 | -2.98 | +3.77 | 59.49% | +33.80% |
| | | | 80% / 20% | | 73.55 | **89.35** | 3.51 | -5.81 | +15.80 | 79.15% | +43.13% |

Table 2: Symmetric vs Asymmetric pruning on ImageNet-100 for ViT-B/16 and DeiT-B/16. Baseline accuracy is shown once per model; pruning results are reported before and after fine-tuning (FT). ΔAcc. denotes the difference between Final and Baseline accuracies; Recovered Acc. denotes the improvement of Final over Pruned.

**LLM Usage:** In preparing this manuscript, we only used large language models (LLMs) to assist with writing flow and polishing the text for clarity and readability. No part of the research design, methodology, experiments, analysis, or results was generated by or dependent on an LLM; all scientific contributions are solely the work of the authors.

**Algorithm 1** HEART-ViT: Head & Token–Aware Dynamic Pruning

---

**Require:** Pretrained ViT $f_\theta$, data distribution $\mathcal{D}$, layers $\ell = 1{:}L$ with tokens $t_j^\ell$ and heads $h_k^\ell$, keep policies $(p_T, p_A)$ *or* loss budget $\varepsilon$, temperature schedule $\gamma(t)$
**Ensure:** Per-input masks $M_T^\ell(x)$, $M_A^\ell(x)$ for inference; optional soft gates $G_z^\ell(x)$ for fine-tuning
    **Phase A: Calibration (one-time or infrequent)**
1: Sample a small calibration batch $\mathcal{B} \sim \mathcal{D}$
2: **for** $\ell = 1$ to $L$ **do**
3:     **for all** $z \in \{t_j^\ell,\, h_k^\ell\}$ **do**
4:         $\bar{S}_z \leftarrow \dfrac{1}{|\mathcal{B}|} \sum\limits_{(x,y)\in\mathcal{B}} \langle \mathcal{H}_z(x)\, z(x),\, z(x) \rangle$         ▷ HVP via Pearlmutter; 2 backprops
5:     **end for**
6:     $\mu_\ell \leftarrow \mathrm{mean}\{\bar{S}_z\}_{z\in\mathcal{C}_\ell}$;   $\sigma_\ell \leftarrow \mathrm{std}\{\bar{S}_z\}_{z\in\mathcal{C}_\ell}$
7: **end for**
8: choose thresholds $\tau_T^\ell, \tau_A^\ell$ by percentiles $(p_T, p_A)$ or by loss budget with $\sum_\ell \varepsilon_\ell = \varepsilon$
    **Phase B: Inference (per-input hard selection)**
9: Forward once to cache activations $\{t_j^\ell(x),\, h_k^\ell(x)\}$
10: **for** $\ell = 1$ to $L$ **do**
11:     **for all** $z \in \{t_j^\ell(x),\, h_k^\ell(x)\}$ **do**
12:         $S_z(x) \leftarrow \langle \mathcal{H}_z(x)\, z(x),\, z(x) \rangle$   ;   $\hat{S}_z(x) \leftarrow \big(S_z(x) - \mu_\ell\big)/\sigma_\ell$
13:     **end for**
14:     **if** percentile policy **then**
15:         Keep top-$p_T$ tokens and top-$p_A$ heads by $S_z(x)$
16:     **else if** loss-budget policy **then**
17:         Select largest $\mathcal{Z}^\ell(x)$ s.t. $\sum_{z\in\mathcal{Z}^\ell(x)} S_z(x) \leq 2\varepsilon_\ell$
18:     **end if**
19:     **for all** $z \in \mathcal{C}_\ell$ **do**
20:         $M_z(x) \leftarrow \mathbb{I}[z \in \mathcal{Z}^\ell(x)]$; apply masks: $t_j^\ell \leftarrow M_{t_j^\ell}(x)\, t_j^\ell$;  $h_k^\ell \leftarrow M_{a_k^\ell}(x)\, h_k^\ell$
21:     **end for**
22: **end for**
23: **return** $f_\theta(x; M)$
    **Phase C: Fine-tuning (optional; soft gates)**
24: **for** training step $t = 1, 2, \dots$ **do**
25:     Sample minibatch $\mathcal{B}$; forward to get activations
26:     **for** $\ell = 1$ to $L$ **do**
27:         Compute $S_z(x)$ and $\hat{S}_z(x)$ as above (optionally on a subset)
28:         $G_z(x) \leftarrow \sigma\big(\gamma(t) \cdot (\hat{S}_z(x) - \tau^\ell)\big)$         ▷ soft gate
29:         Apply gated forward: $t_j^\ell \leftarrow G_{t_j^\ell}(x)\, t_j^\ell$;  $h_k^\ell \leftarrow G_{a_k^\ell}(x)\, h_k^\ell$
30:     **end for**
31:     Backprop, update $\theta$; anneal $\gamma(t){\uparrow}$ so $G \to M$
32: **end for**

---

| Dataset | Model | Pruning Strategy | Ratio (Tokens/Heads) | Baseline Acc. | Pruned Acc. (Before FT) | Final Acc. (After FT) | FLOPs (G) | ΔAcc. | Recovered Acc. | FLOPs↓ | Throughput↑ |
|---|---|---|---|---|---|---|---|---|---|---|---|
| ImageNet-1K | ViT-B/16 | Symmetric | 20% / 20% | | 79.14 | **85.20** | 13.45 | +3.74 | +6.06 | 20.16% | +17.86% |
| | | | 40% / 40% | | 77.66 | **84.60** | 10.14 | +3.14 | +6.94 | 39.82% | +22.53% |
| | | | 50% / 50% | | 75.30 | **82.99** | 8.52 | +1.53 | +7.69 | 49.40% | +25.19% |
| | | | 60% / 60% | | 68.72 | **81.35** | 6.83 | -0.11 | +12.63 | 59.48% | +36.01% |
| | | | 80% / 80% | 81.46 | 42.17 | **69.84** | 3.51 | -11.62 | +27.67 | 79.14% | +46.14% |
| | | Asymmetric | 20% / 80% | | 79.93 | **86.17** | 13.45 | +4.71 | +6.24 | 20.16% | +17.68% |
| | | | 40% / 60% | | 76.92 | **84.36** | 10.14 | +2.90 | +7.44 | 39.82% | +22.01% |
| | | | 60% / 40% | | 67.02 | **80.98** | 6.83 | -0.48 | +13.96 | 59.48% | +36.00% |
| | | | 80% / 20% | | 50.54 | **75.94** | 3.51 | -5.52 | +25.40 | 79.14% | +45.71% |
| ImageNet-1K | DeiT-B/16 | Symmetric | 20% / 20% | | 80.7 | **85.08** | 13.45 | +3.28 | +4.38 | 20.16% | +15.81% |
| | | | 40% / 40% | | 77.12 | **84.05** | 10.14 | +2.25 | +6.93 | 39.82% | +21.25% |
| | | | 50% / 50% | | 74.87 | **83.81** | 8.52 | +2.01 | +8.94 | 49.40% | +24.33% |
| | | | 60% / 60% | | 73.79 | **81.43** | 6.83 | -0.37 | +7.64 | 59.48% | +34.28% |
| | | | 80% / 80% | 81.8 | 56.08 | **71.94** | 3.51 | | +15.86 | 79.14% | +44.38% |
| | | Asymmetric | 20% / 80% | | 80.73 | **85.32** | 13.45 | +3.52 | +4.59 | 20.16% | +16.13% |
| | | | 40% / 60% | | 76.00 | **82.45** | 10.14 | +0.65 | +6.45 | 39.82% | +21.29% |
| | | | 60% / 40% | | 65.71 | **77.92** | 6.83 | -3.88 | +12.21 | 59.48% | +34.69% |
| | | | 80% / 20% | | 40.01 | **72.81** | 3.51 | -8.99 | +32.80 | 79.14% | +43.68% |

Table 3: Symmetric vs Asymmetric pruning on ImageNet-1K for ViT-B/16 and DeiT-B/16. Baseline accuracy is shown once per block; pruning results are reported before and after fine-tuning (FT). ΔAcc. denotes the difference between Final and Baseline accuracies; Recovered Acc. denotes the improvement of Final over Pruned.

| Dataset | Model | Pruning Strategy | Ratio (Tokens/Heads) | Baseline Lat. (ms) | Pruned Lat. (Before FT) | Final Lat. (After FT) | Latency Improved (%) |
|---|---|---|---|---|---|---|---|
| ImageNet-Full | ViT-B/16 | Symmetric | 20% / 20% | | 18.78 | 18.08 | 23.1% |
| | | | 40% / 40% | | 14.05 | 14.40 | 38.7% |
| | | | 50% / 50% | | 15.89 | 16.01 | 31.9% |
| | | | 60% / 60% | | 14.79 | 14.47 | 38.5% |
| | | | 80% / 80% | 23.50 | 14.20 | 14.28 | 39.2% |
| | | Asymmetric | 20% / 80% | | 18.73 | 18.59 | 20.9% |
| | | | 40% / 60% | | 14.00 | 13.95 | 40.6% |
| | | | 60% / 40% | | 15.59 | 14.57 | 38.0% |
| | | | 80% / 20% | | 16.88 | 14.45 | 38.5% |
| | DeiT-B/16 | Symmetric | 20% / 20% | | 18.12 | 18.11 | 23.9% |
| | | | 40% / 40% | | 14.86 | 15.18 | 36.2% |
| | | | 50% / 50% | | 16.20 | 15.84 | 33.4% |
| | | | 60% / 60% | | 15.56 | 15.44 | 35.1% |
| | | | 80% / 80% | 23.78 | 14.76 | 15.49 | 34.9% |
| | | Asymmetric | 20% / 80% | | 18.31 | 18.22 | 23.4% |
| | | | 40% / 60% | | 14.45 | 14.36 | 39.6% |
| | | | 60% / 40% | | 15.22 | 16.20 | 31.9% |
| | | | 80% / 20% | | 15.40 | 14.59 | 38.7% |

Table 4: AGX Orin edge device latency analysis for DeiT-B/16 and ViT-B/16 under Symmetric vs Asymmetric pruning on ImageNet-1k. Latency improvement is computed as $(\text{Baseline} - \text{Final})/\text{Baseline} \times 100$. Higher values indicate larger speedups.

| Dataset | Model | Pruning Strategy | Ratio (Tokens/Heads) | Baseline Lat. (ms) | Pruned Lat. (Before FT) | Final Lat. (After FT) | Latency Improved (%) |
|---|---|---|---|---|---|---|---|
| ImageNet-100 | DeiT-B/16 | Symmetric | 20% / 20% | | 18.04 | 18.03 | 21.8% |
| | | | 40% / 40% | | 14.06 | 13.62 | 40.9% |
| | | | 50% / 50% | | 17.02 | 15.82 | 31.3% |
| | | | 60% / 60% | | 15.49 | 14.10 | 38.8% |
| | | | 80% / 80% | 23.05 | 14.73 | 13.68 | 40.6% |
| | | Asymmetric | 20% / 80% | | 19.38 | 18.74 | 18.7% |
| | | | 40% / 60% | | 14.41 | 13.59 | 41.0% |
| | | | 60% / 40% | | 16.67 | 14.99 | 34.9% |
| | | | 80% / 20% | | 14.84 | 13.72 | 40.5% |
| | ViT-B/16 | Symmetric | 20% / 20% | | 18.13 | 18.07 | 22.9% |
| | | | 40% / 40% | | 13.61 | 13.58 | 42.0% |
| | | | 50% / 50% | | 15.21 | 15.07 | 35.7% |
| | | | 60% / 60% | | 14.06 | 13.34 | 43.1% |
| | | | 80% / 80% | 23.43 | 13.99 | 13.72 | 41.4% |
| | | Asymmetric | 20% / 80% | | 19.47 | 18.97 | 19.1% |
| | | | 40% / 60% | | 14.04 | 13.62 | 41.9% |
| | | | 60% / 40% | | 14.41 | 13.53 | 42.3% |
| | | | 80% / 20% | | 14.05 | 13.62 | 41.9% |

Table 5: AGX Orin edge device latency latency analysis for DeiT-B/16 and ViT-B/16 under Symmetric vs Asymmetric pruning on ImageNet-100. Latency improvement is computed as $(\text{Baseline} - \text{Final})/\text{Baseline} \times 100$. Higher values indicate larger speedups.

| Method | FLOPs (G) | Top-1 Acc. (%) | Category |
|---|---|---|---|
| ViT-Base/16 Dosovitskiy et al. (2021) | 17.6 | 77.9 | Baseline |
| DeiT-Base/16 Touvron et al. (2021) | 17.6 | 81.8 | Baseline |
| CrossViT-B Chen et al. (2021) | 21.2 | 82.2 | SOTA |
| T2T-ViT-24 Yuan et al. (2021) | 14.1 | 82.3 | SOTA |
| TNT-B Han et al. (2021) | 14.1 | 82.8 | SOTA |
| Swin-B Liu et al. (2021) | 15.4 | 83.3 | SOTA |
| LV-ViT-M Jiang et al. (2021) | 12.7 | 84.0 | SOTA |
| DynamicViT-LV-M/0.8 Rao et al. (2021) | 9.6 | 83.9 | SOTA |
| **Our Results (ViT-B/16)** | | | |
| *Symmetric Pruning* | | | |
| Sym 20/20 | 13.45 | 85.20 | Ours |
| Sym 40/40 | 10.14 | 84.60 | Ours |
| Sym 50/50 | 8.52 | 82.99 | Ours |
| Sym 60/60 | 6.83 | 81.35 | Ours |
| Sym 80/80 | 3.51 | 69.84 | Ours |
| *Asymmetric Pruning* | | | |
| Asym 20/80 | 13.45 | 86.17 | Ours |
| Asym 40/60 | 10.14 | 84.36 | Ours |
| Asym 60/40 | 6.83 | 80.98 | Ours |
| Asym 80/20 | 3.51 | 75.94 | Ours |
| **Our Results (DeiT-B/16)** | | | |
| *Symmetric Pruning* | | | |
| Sym 20/20 | 13.45 | 85.08 | Ours |
| Sym 40/40 | 10.14 | 84.05 | Ours |
| Sym 50/50 | 8.52 | 83.81 | Ours |
| Sym 60/60 | 6.83 | 81.43 | Ours |
| Sym 80/80 | 3.51 | 71.94 | Ours |
| *Asymmetric Pruning* | | | |
| Asym 20/80 | 13.45 | 85.32 | Ours |
| Asym 40/60 | 10.14 | 82.45 | Ours |
| Asym 60/40 | 6.83 | 77.92 | Ours |
| Asym 80/20 | 3.51 | 72.81 | Ours |

Table 6: Comparison of our pruning results with state-of-the-art (SOTA) vision transformer variants on ImageNet-1K. We report FLOPs (billions, G) and Top-1 accuracy. Our method achieves superior trade-offs between accuracy and efficiency in both symmetric (Sym) and asymmetric (Asym) pruning settings.

| Device | GPU Cores | Max GPU Frequency | Shared Memory |
|---|---|---|---|
| AGX Orin | 1792 | 930 MHz | 32 GB |
| Orin NX | 1024 | 918 MHz | 16 GB |
| Orin Nano | 512 | 625 MHz | 4 GB |
| AGX Orin Devkit (old version) | 2048 | 1.3 GHz | 32 GB |
| Orin NX | 1024 | 765 MHz | 8 GB |
| Orin Nano | 1024 | 625 MHz | 8 GB |

Table 7: Specifications of Various Orin Devices

