# OpenReview forum: "HEART-ViT: HESSIAN-GUIDED EFFICIENT DYNAMIC ATTENTION AND TOKEN PRUNING IN VISION TRANSFORMERS"
_ICLR.cc/2026/Conference — ICLR 2026 Conference Withdrawn Submission_

### Official Review · Reviewer_hz49 · 2025-10-29

**Soundness:** 3
**Presentation:** 1
**Contribution:** 2
**Rating:** 2
**Confidence:** 5

**Summary:**

This paper proposes a hessian-guided ViT pruning technique, which is applied on both tokens and attention heads. The biggest strength of the paper is its mathematical rigor, as the motivation, formulation and solution of the problem all followed a clear math process, providing a strong insight into the method's working mechanism. However, there are fundamental flaws in the paper's experiments, for lack of comprehensive comparison, and poor presentation, which renders the paper below the standard for an ICLR paper.

**Strengths:**

- Very clear math behind the proposed method, all the way through the motivation, the formulation and the solution
- Unified token and attention head pruning provides a novel angle for ViT pruning
- Experimental results are competitive against vanilla baselines
- The method supports both finetuned and non-finetuned modes

**Weaknesses:**

- Overall, the quality of presentation is not great – this includes inconsistent spacing (eg. Ln226,231), lack of proper explanation of symbols (ln178), poor quality of figures (e.g. fig 1,2,3). The authors are encouraged to work on the improvement of presentation vigorously to match top-tier conference standards.
- The biggest weakness of the paper is the experiment section. It generally lacks the comprehensive evaluations required by a solid study for a new efficient ViT method. The range of SotA baseline methods is lacking, so is the variety in model size is lacking, and the diversity in the downstream tasks (OD, seg, etc), and the coverage for datasets is limited (only ImageNet)

**Questions:**

- I recommend removing Table 1 for something that provides more information that is useful to the audience from this area. Table 1 is too verbose and contains many items that are very technique-specific.
- Figure 1 and Figure 2 are good for teasers but poor choices for presenting as main results - table with plain numbers are easier to read for the main results. Comparisons are clearer too with tables.

---

### Official Review · Reviewer_grLo · 2025-10-29

**Soundness:** 4
**Presentation:** 2
**Contribution:** 2
**Rating:** 2
**Confidence:** 5

**Summary:**

This paper proposes HEART-ViT, a joint token and head pruning method for efficient ViTs. HEART-ViT employs a second-order Taylor expansion on a converged ViT model to approximate the loss change induced by pruning. The resulting scoring criterion is simplified to rely solely on the curvature term, enabling efficient computation. The proposed method is evaluated on ViT-B and DeiT-B backbones, demonstrating good performance and efficiency.

**Strengths:**

1. The mathematical motivation is solid. The proposed scoring strategy is sound.

2. The experiments on backbone model are comprehensive.

3. This work includes evaluation on edge devices, further demonstrating its practical effectiveness.

**Weaknesses:**

1. __Insufficient empirical results:__ This paper lacks a bunch of experiments:

   * Comparisons to prior token and/or structural pruning methods;
   * Performance on different ViT architectures (e.g., Swin Transformer) and sizes (e.g., ViT/DeiT-Small)
   * Performance on downstream tasks after pruning

   Although the analytical studies on latency, pruning effectiveness, and layerwise similarity are informative, the absence of these fundamental experiments significantly undermines the empirical strength and overall significance of the work.

2. __Massive finetuning demands:__ HEART-ViT requires 100-epoch finetuning to be effective, which is an essential drawback compared to state-of-the-art token pruning/merging methods that usually require a few finetune epochs or none at all. This substantially reduces the practical efficiency and applicability of HEART-ViT.

3. __Poor presentation:__ The presentation quality is low:

   * Citation format does not align with the ICLR format
   * Overlapping between Figure 6 and Figure 5's caption
   * Overlapping between some formulae and surrounding texts
   * Some formulae have equation numbers while some others do not
   * Figure 1 is never referred to
   * Inconsistent reference formats (btw, ToMe should be published on ICLR but labelled CVPR)

**Questions:**

Please refer to the weaknesses above

---

### Official Review · Reviewer_oymU · 2025-10-30

**Soundness:** 3
**Presentation:** 4
**Contribution:** 3
**Rating:** 2
**Confidence:** 5

**Summary:**

This paper introduces a unified token and head pruning algorithm for ViTs called HEART-ViT. HEART-ViT measures the loss perturbation caused by removing certain tokens or attention heads to identify redundant components. It also proposes a simplified formulation based on the second-order Taylor expansion of the converged model for a efficient measurement implementation. Experiments on different ViT backbones demonstrate its effectiveness.

**Strengths:**

* The unified form of token and head pruning for ViT with mathematical proof is novel and interesting.

* The visualizations effectively illustrate the pruning behavior and provide valuable interpretability.

* Broad studies on model performance with different pruning ratios.

**Weaknesses:**

* No comparisons are provided with state-of-the-art efficient ViT methods, particularly recent token or structural pruning and merging approaches [1,2,3,4].

* The experiments are limited to ViT-B and DeiT-B, which share nearly identical architectures, leaving the method’s generalization to other ViT variants unverified.

* The comparison with the backbone in Appendix Tables 3&4 is unfair, as the baseline is not finetuned for the same 100 epochs used by the proposed method.

* The contribution appears marginal compared to AdaViT [5], which also unifies token, head, and block pruning while employing a simpler and more practical estimation strategy.

[1] Bolya, Daniel, et al. "Token merging: Your vit but faster." ICLR, 2023.

[2] Yang, Huanrui, et al. "Global vision transformer pruning with hessian-aware saliency." CVPR, 2023.

[3] Kim, Minchul, et al. "Token fusion: Bridging the gap between token pruning and token merging." WACV, 2024.

[4] Wang, Hongjie, Bhishma Dedhia, and Niraj K. Jha. "Zero-TPrune: Zero-shot token pruning through leveraging of the attention graph in pre-trained transformers." CVPR, 2024.

[5] Meng, Lingchen, et al. "Adavit: Adaptive vision transformers for efficient image recognition." CVPR, 2022.

**Questions:**

* How do you explain the performance different between symmertic and asymmertic pruning? Asymmetric pruning seems to be more flexible but introduces worse trade-off bwteen efficiency and performance in most cases. Does ViT architecture prefer a certain type of pruning?

* How do you explain the significant performance degradation after pruning pre-finetuning? Notably, many state-of-the-art methods are finetuning-free.

* Why do you specifically choose second-order Taylor expansion rather than more accurate k-th Taylor expansion?

---

### Official Review · Reviewer_8ymd · 2025-11-09

**Soundness:** 3
**Presentation:** 3
**Contribution:** 3
**Rating:** 4
**Confidence:** 4

**Summary:**

This paper proposed a Hessian-guided pruning framework for ViTs, which jointly pruning both tokens and attention heads based on second-order sensitivity. The second order information is gathered using Hessian–vector products (HVPs) method, for each token and head.
Empirical evaluations on ImageNet on ViT and DeiT show up to 49% FLOPs reduction, 36–43% latency improvements. Hardware benchmarking also shows realistic efficiency improvement.

**Strengths:**

The second-order based pruning criterion is well motivated.
The authors also made endeavors to test on real world edge device to evaluate the hardware efficiency.

**Weaknesses:**

Although authors mentioned the complexity analysis of the HVP calculation, the level of empirical overhead is still concerning as it requires 2 backward passes.
Also the paper lacks discussion on their differences and advances against existing hessian-based ViT pruning papers, e.g. LPViT [1], NViT [2].

**Questions:**

1. There are too little comparisons with SOTA methods, especially missing more recent papers that also adopts hessian in pruning, e.g. LPViT [1] (ECCV'24), NViT [2] (CVPR'23).
2. Although authors provided hardware benchmarking results, i wonder how is it compared with SOTA methods.

Minor problems:
Figure 1 there are some visual clarity issue.


Reference:
[1] Xu, Kaixin, et al. "Lpvit: Low-power semi-structured pruning for vision transformers." European Conference on Computer Vision. Cham: Springer Nature Switzerland, 2024.
[2] Yang, Huanrui, et al. "Global vision transformer pruning with hessian-aware saliency." Proceedings of the IEEE/CVF conference on computer vision and pattern recognition. 2023.

---

### Note · Authors · 2025-12-03

I have read and agree with the venue's withdrawal policy on behalf of myself and my co-authors.